# Frequency of Injury and Illness in the Final 4 Weeks before a Trail Running Competition

**DOI:** 10.3390/ijerph18105431

**Published:** 2021-05-19

**Authors:** Rubén Gajardo-Burgos, Manuel Monrroy-Uarac, René Mauricio Barría-Pailaquilén, Yessenia Norambuena-Noches, Dina Christa Janse van Rensburg, Claudio Bascour-Sandoval, Manuela Besomi

**Affiliations:** 1Facultad de Medicina, Instituto de Aparato Locomotor y Rehabilitación, Universidad Austral de Chile, Valdivia 5090000, Chile; manuelmonrroy@uach.cl (M.M.-U.); rbarria@uach.cl (R.M.B.-P.); 2Instituto de Enfermería, Universidad Austral de Chile, Valdivia 5090000, Chile; 3Escuela de Kinesiología, Facultad de Medicina, Universidad Austral de Chile, Valdivia 5090000, Chile; norambuena.y@gmail.com; 4Section Sports Medicine, Faculty of Health Sciences, University of Pretoria, Pretoria 0028, South Africa; christa.jansevanrensburg@up.ac.za; 5Medical Board Member, International Netball Federation, Manchester M1 5LN, UK; 6Departamento de Medicina Interna, Universidad de La Frontera, Temuco 4781218, Chile; claudio.bascour@ufrontera.cl; 7Carrera de Kinesiología, Facultad de Ciencias de la Salud, Universidad Autónoma de Chile, Temuco 4810101, Chile; 8Escuela de Kinesiología, Universidad del Desarrollo, Santiago 7610658, Chile; mbesomim@udd.cl

**Keywords:** athletic injuries, musculoskeletal injuries, disease, injury management, trail running

## Abstract

We aimed to (i) determine self-reported injury and illness frequency in trail runners 4 weeks preceding competition; (ii) compare athletes with and without injury/illness by sex, age, body mass index (BMI) and competition distance; (iii) describe mechanism of injury, anatomical region (injury)/organ system (illness) involved, consequences of injury on preparation and self-perception of injury severity; (iv) compare anatomical region (injury) and organ system (illness) by sex. A total of 654 trail runners (age 36.2, IQR 30.6–43.0; 36.9% females) participated in this retrospective cross-sectional study by completing a self-reported questionnaire. Injury and illness frequency rates were 31.3% (*n* = 205, CI: 27.7–35.0%) and 22.3% (*n* = 146, CI: 19.1–25.7%), respectively. No significant difference was found between injured vs. non-injured or ill vs. non-ill study participants by sex, age, BMI and competition distance. Regarding injuries, gradual onset (41.6%) and knee (33.2%) were the most indicated mechanism and anatomical region of injury. At least 85.4% of trail runners changed their training following injury and 79% indicated that their injury would affect their competition performance. Regarding illness, the respiratory tract was the most frequent organ system involved (82.9%). Male and female participants reported similar proportions of anatomical regions (injury) and organ systems (illness) affected. These results could help to generate education strategies and appropriate medical support before and during these competitions.

## 1. Introduction

Running is one of the most popular forms of physical activity because of its accessibility and low cost required for participation. Positive lifestyle changes including physical, psychological and social benefits are attributed to running [1]. Trail running is one of the running disciplines that has grown worldwide over the last years [2]. According to the International Trail Running Association, trail running is a pedestrian race, in a natural environment (e.g., mountain, forest, prairie) containing a minimal of asphalted or paved routes (under 20% of the total route) and which ideally, though not necessarily should be self-sufficient (i.e., the runner must be autonomous between helping or supplying stations regarding clothing, communication, food and drinks) [3].

Despite all the health benefits associated with physical activity, the risk of medical encounters such as injuries and illness is relatively high. The incidence of running-related injuries in disciplines covering from track and field athletes to marathon runners ranges from 19.4 to 79.3% [4] over twelve weeks to one year of follow-up. Kluitenberg et al. [5] described the prevalence of injuries in different populations of runners depending on their experience and type of competition, reporting a prevalence of 16.7 to 84.9% in novices, 21.6 to 55.0% in recreational runners, 1.4 to 94.4% in cross country runners and 21.0 to 90.6% in ultra-distance runners [5]. A recently published systematic review [6] showed an overall incidence range between 1.6–4285.0 injuries per 1000 h of running. Additionally, the prevalence of injuries in trail runners has been reported between 22.2% [7] during competition and 28.2% [8] over one year with the lower limbs the most susceptible to sustain injuries. Illnesses are also prevalent among athletes. Viljoen et al. [6] showed an overall incidence range between 65.0–6676.6 illnesses per 1000 h of running and the digestive system was the most common trail running illness reported. Moreover, upper respiratory illness that interfere with training and performance during competition has been reported in up to 10% of athletes [9]. Other systems commonly affected by illness are skin and subcutaneous tissues and the genitourinary system [10]. These injuries and illnesses have important consequences for the athletes’ health, such as early-onset osteoarthritis in the long term, rising health costs [11], absenteeism from work, loss in training days, diminished performance in the short term and, in some cases, dropping out of sport participation [12] and a negative impact on performance [9].

Injuries related to trail running are multifactorial and generally described as gradual onset injuries (repetitive micro-trauma) and sudden onset injuries (muscle strain, ligament sprains, etc.) [13]. Bertelsen et al. [14] present an evidence-informed conceptual framework outlining the multifactorial nature of running-related injury etiology. This framework hypothesizes that a running-related injury occurs when accumulated loads per running session exceed the structure-specific load capacity when entering a running session. The structure-specific load capacity when entering a running session can adapt positively or negatively over time. This ability to adapt can be influenced by factors such as sex, age, or running distance. Loads associated with running could be quantified by the number of steps and the magnitude of the load in each step, so variables such as running distance, weight and BMI are also relevant to analyze. Some factors commonly associated with running injuries are the history of previous injuries and training programs with excessive overload [15]. Training load changes during the pre-competitive phase of an athlete’s training plan. The volume and intensity of training increase and then progressively decrease before the competition, a phenomenon known as tapering [16]. This increase in training load could affect the athlete’s stress levels, exposing them to a greater injury risk [17] and increases the risk of illness due to alterations to the immune system [9]. Athletes are particularly vulnerable during the season of major competitions when they continue their training despite having an injury or illness [15,18,19].

A four-step strategy for injury prevention has been proposed by van Mechelen et al. [20]: (i) to know the extent of the problem; (ii) to know the risk factors and injury mechanisms; (iii) to introduce a preventive measure; and (iv) to assess the effectiveness of this preventive measure. From the perspective of knowing the extent of the problem, studying the prevalence of injuries around a competition phase may be important to understand the magnitude of the problem [11,15,18,19,21], with the ultimate goal to generate education strategies and provide better medical support during competitions. Only a few studies have determined the frequency or prevalence of injuries or illnesses in trail runners [8,11,13,22,23] and no available studies have focused on the specific stage before the competition. This study aimed to (i) determine the self-reported frequency of injuries and illnesses in athletes participating in a trail running competition over the four weeks preceding the competition; (ii) compare athletes with and without injury/illness by sex, age, body mass index (BMI) and competition distance; (iii) describe the mechanism of injury, the anatomical region (injury) or organ system (illness), consequences of injury on training features (i.e., volume, intensity and performance) and self-perception of the severity of injury for competition; and (iv) compare the anatomical region among injured athletes and the system affected among ill athletes by sex.

## 2. Materials and Methods

### 2.1. Study Design and Setting

The study used a retrospective cross-sectional design. A total number of 1185 athletes entered the trail running event that occurred on 24 and 25 June 2017. The competition distances included continuous race distances over 11, 18, 24, 45 and 63 km and a two-phase distance Travesía Crossing of 23 km and 22 km over two days. During the reception of the competition kit one day before the event, all athletes were personally invited to participate.

### 2.2. Participants

Competition entrants 18 years and older that registered for the event and signed a participation informed consent were included in the study. All information was treated as strictly confidential and the injury and illness reports were anonymously recorded according to the guidelines of the Declaration of Helsinki. This research was assessed and approved by the Ethics Committee of the Medicine Faculty of Austral University of Chile (3003.2017) and followed the Strengthening the Reporting of Observational Studies in Epidemiology (STROBE) Guidelines.

### 2.3. Data Sources and Variables

A self-reported questionnaire in hard copy was completed the day before the competition by all participants who agreed to participate in the study, regardless of if they competed or not. They were assisted by physiotherapists previously trained in the completion of the questionnaire. The questionnaire was based on preceding surveys used by the International Association of Athletics Federations and the Fédération Internationale de Natation during the World Championships in 2013 [15,19]. We used a forward translation and backward verification method to translate the English version into Spanish, adjusting the questions to trail runners. This process involved the translation of the questionnaire by a bilingual translator with a Spanish mother tongue, followed by the verification of the translated version by a bilingual person with an English mother tongue [24]. The questionnaire was divided into two sections. In the first section, questions covered sociodemographic data (age, sex, height and weight), distance of competition, weekly training hours in the last 4 weeks and questions to determine health status (injuries and illnesses in the last 4 weeks). Injury was defined as “injuries or physical complaints (such as pain, ache, stiffness, swelling, instability/giving way, locking or other symptoms) that athletes had in the four weeks prior to the championship, even if this had no major consequences for the athlete’s participation in normal training and/or competition” [25]. Illnesses were defined as “any symptoms implying infection, allergy, gastroenteritis, flu, or dehydration that affected the competition preparation”. If the athletes did not report any injuries or illnesses in the 4 week period before the competition, the questionnaire was ended.

The second section (if athletes reported an injury or illness) was composed of questions about injury mechanism, anatomical regions of injury, self-perceived impact of injury on training preparation for the event (i.e., volume, intensity and performance), self-perceived impact of injury on the competition (i.e., severity) and organ system affected by illness.

### 2.4. Statistical Methods

The Shapiro–Wilk test was used to analyze whether quantitative variables had a normal distribution. Since this assumption was not reached by any variable, median and interquartile range (IQR) were reported. A 95% confidence interval (CI) was used in the frequency analyses. Missing data were ignored (complete case method) and we reported the extent of missing data. In order to assess the external validity of our findings, the study participants and registered athletes in the event were compared using chi-square (sex and competition distance) and Mann–Whitney U tests (age). For aim 1, descriptive statistics were used to describe the frequency of injury and illness among athletes. For aim 2, injured and non-injured athletes and ill and non-ill athletes were compared by age and BMI using the Mann–Whitney test and by sex and competition distance using the chi-square test. For aim 3, descriptive statistics were used. For aim 4, anatomical regions and organ systems were compared within injured/ill athletes by sex (males/females) using the chi/square test. Data were analyzed using STATA 14 software.

## 3. Results

Of the 1185 registered entrants, 654 (55.2%) runners, 241 (36.9%) females and 413 (63.1%) males accepted to participate and completed the questionnaire. There were no significant differences between the group of participants and total competition entrants regarding sex (*p* = 0.546), age (*p* = 0.508) and competition distance (*p* = 0.100).

### 3.1. Aim 1: Frequency of Injury and Illness

Regarding injuries, 31.3% (*n* = 205, CI: 27.7–35.0%) of participants indicated they had an injury in the 4 weeks prior to the competition. Of those who reported injuries, 62 (30.2%) reported having more than one injury during the previous 4 weeks.

Illness during the 4 weeks before competition was reported by 22.3% of participants (*n* = 146, CI: 19.1–25.7%). The frequency of injuries and illnesses in the 4 weeks before the event is shown in Table 1.

### 3.2. Aim 2: Comparison of Injured and Ill Athletes by Sex, Age, BMI and Competition Distance

There was no statistically significant difference between injured vs. non-injured or ill vs. non-ill athletes by sex, age, BMI, or competition distance (Table 1).

### 3.3. Aim 3: Description of the Mechanism of Injury, Anatomical Region (Injury) or Organ System (Illness), Consequences of Injury on Training (i.e., Volume, Intensity and Performance) and Self-Perception of the Severity of Injury for Competition

Of the 205 athletes reporting an injury, most athletes (41.6%; *n* = 84) indicated that the injury had gradual onset, 26.7% (*n* = 54) reported sudden onset without trauma, 20.8% (*n* = 42) reported an identifiable event such as a fall while running and 10.9% (*n* = 22) an incident unrelated to running.

Nine participants (4 females and 5 males, 4.4%) did not report the injured anatomical region. The knee was the most affected site (33.2%) for both sexes, followed by the hips in females (14.3%) and the ankle in males (16.7%) (Figure 1). From the reported illnesses (*n* = 146), the respiratory system was the most frequent organ involved (*n* = 121, 82.9%), followed by the gastrointestinal system (*n* = 10, 6.9%), the immunological system (*n* = 7, 4.8%) and the musculoskeletal system (*n* = 3, 2.1%).

Among participants who reported a previous injury, 77.6% had to reduce their training volume as a consequence of the injury, 80.3% had to modify the intensity of their training and 85.4% found their performance in training was affected by their injuries (Table 2). Participants had to modify their training within the last 4 weeks because of these injuries by a median of six days (IQR 2–12 days).

Concerning self-perception of injury severity and its effect on performance during the competition, 21.0% did not believe it would have any influence, whereas 40.5% indicated it would have a mild effect, 30.0% believe it would have a moderate effect, 5.0% believe it would have a significant effect and 3.5% declared that they would not take part in the competition since they considered that their injury would have a serious effect.

Regarding mechanism of injury and anatomical region of injury between runners with and without time loss injuries, we found that gradual onset injuries was the most frequent for both groups (38.4% vs. 51.0%) and that the most affected anatomical region was the knee (35.9% vs. 25.5%). Observing the self-perception of the severity of the injury for the competition, the runners with time loss injuries believed that their injury would mainly affect them mildly (40.7%) or moderately (34.0%). Runners with non-time loss injuries believed that this injury would mainly affect them mildly (40.0%) or not at all (38.0%).

### 3.4. Aim 4: Comparison of the Anatomical Region (Injury) and the Organ System (Illness) Affected among Athletes by Sex

Male and female participants reported similar proportions of anatomical regions affected by injury (*p* = 0.062) and organ systems affected by illness (*p* = 0.211) (Figure 1).

## 4. Discussion

Knowledge of injuries and illness in trail runners, as well as knowledge of the factors that could influence in structure-specific load capacity when entering a running session and the factors that could influence the magnitude of load associated with running are important for the understanding of the etiology of injuries and illness [14]. This study aimed to answer several questions. First, our results show that nearly a third of trail runners compete with some type of injury and almost a quarter competes with an illness sustained in the four weeks before the competition. Second, there were no differences in age, sex, BMI, or competition distance between injured vs. non-injured, or ill vs. healthy participants. Third, one in four athletes reported a gradual onset of injuries, one in three reported injuries at the knee and four in five illnesses to the respiratory system. Fourth, there were no differences between male and female participants regarding the anatomical region of injuries or the system affected by illnesses.

Participation in sports events despite illness or injury is a known phenomenon in athletes. Studies conducted in athletes from national teams participating in International Championships of track and field athletics reported that 24.3% to 29.2% of them had suffered some type of injury and 17.6% of them some illness, during the 4 weeks before a competition [15,18]. Although the number of runners participating in recreational/competitive events has highly increased during the last decades [2], there is limited information that determines the condition of these athletes when entering into these events. Records obtained from elite athletes are not comparable to recreational/competitive athletes mainly due to multiple differences presented during their training, heterogeneity in their physical conditions and differences in health insurance [5]. A recently published study on trail runners found that 28.2% of them suffered from an injury or illness in the 12 months leading up to a competition [8]. The current study is the first study that determines the prevalence of injuries and illnesses in the final 4 weeks before the competition in trail runners. To our knowledge, only two studies have analyzed the prevalence of injuries prior to a competition. Viljoen et al. concluded that 28.2% of the participants of a trail running competition reported an injury in the twelve months before a competition [8]. In a prospective study, an average prevalence of 22.4% (95% CI: 20.9–24.0%) was reported in trail runners over a six month follow-up period [11]. As we evaluated a specific period in the 4 weeks pre-competition our results are not directly comparable.

The frequency of injuries (31.3%) in the 4 weeks before the competition appears high considering that one of the main risk factors to suffer an injury is to report a previous injury [7,19] and that these athletes aim to get to the competition at their best performance level. The prevalence of injury in runners ranges from 0.5 to 94.4% [4,5]. This variation is mainly due to dissimilarities in the definition of injury, type of athletes and different monitoring periods.

Similar to injuries, illnesses in the 4 weeks before a competition may generate alterations in the quantity and quality of training in athletes, as well as affect competition performance and put the athlete at greater health risk during competition [26]. In our study, 22.3% showed signs and symptoms of illnesses in the 4 weeks before competition. This finding is consistent with what was found by Edouard et al. in athletes participating in the European Athletics Indoor Championships [18]. Over 80% of these illnesses were associated with the respiratory system. It has been well documented that illnesses of the upper respiratory tract are the most prevalent in different sports populations [9]. Most cases are related to common viral infections and allergic responses [9]. A recently published systematic review [6] showed that the most common trail running illnesses reported related to the gastrointestinal tract, followed by the metabolic and cardiovascular systems. These illnesses were only analyzed during competition but not during the training period leading up to the competition. It has been suggested that long periods of high intensity of physical exercise or an increase of load in preparation for a competition, increases the risk of alterations to the immune system and therefore contraction of an illness [9,10]. In our case, considering the timing of assessment was around winter, this high prevalence was expected.

There were no differences in sex, age, BMI or competition distance between athletes that reported an injury/illness and those who did not. Regarding the comparison by sex, a recent systematic review found no differences between the overall injury rate between females and males [20.8 per 100 (95% CI 19.9–21.7) vs. (20.4 (95% CI 19.7–21.1) respectively] [27]. Observing competition distance, a higher incidence of injury in females competing in 10 kilometres or less (RR of 1.08; 95% CI: 1.04–1.39) was the only finding, but the analysis did not incorporate trail running studies [27]. There is no consensus on the association between age and risk of injury in runners. A systematic review [28] found that older runners were at higher risk, but this relationship was only investigated in one of the eight articles that were included. Caution with interpretation is required as the study analysed runners preparing for a marathon and not a trail running competition. Regarding the BMI, a systematic review [29] found that higher BMI was a risk factor of injury in short distance runners (mean running distance ≤20 km/week and ≤10 km/session) but this analysis did not incorporate studies with trail runners.

Forty-two percent of injured athletes declared that their injury started gradually. This is in line with previous studies reporting that gradual onset injuries are the most common mechanism of injury in trail runners [8,11]. Overuse (gradual onset) injuries generally do not generate a loss of training/competition days, but they do generate alterations in the performance during those days [30]. This type of injury could be related to inappropriate planning of training load and competition; including an excessive and rapid increase in volume and intensity, such as those that occur in the pre-competition phase [16]. This situation usually does not allow time for correct adaptation of tissues that need to recover [17].

As reported in previous studies [4,5,8], the most commonly affected site was the knee (33.2%). In our population, this may be linked, on the one hand, to the high demands and impact loads required for downhill running which demand higher muscle co-activation to improve joint stiffness [23]. On the other hand, a hilly run could generate fatigue of the plantar flexor muscles [31], exposing runners to a lower capacity for postural control, which could explain the higher frequency of lower limb injuries. However, this warrants further evaluation.

Training and competition generate a series of homeostatic changes, which produce different types of positive adaptations in the human body [32]. At least 85.4% of trial runners needed to change their training because of the injury. This change or unintended decrease of training may generate a reduction in performance [33], cause fear and anxiety [34] and increase injury risk [32]. In order to determine the severity of the injury we have used self-perception of the effect of this injury on running performance during the competition. More than three-quarters of participants reported that their injury would affect their running performance. These beliefs could cause stress and anxiety in runners [34], which could further negatively influence performance at the competition itself. Other studies have used the OSTRC-H severity score to determine the severity of the injury, analysing the consequences in training resulting from the injury. Severity scores ranged between 31.6 and 35.0 on a 0 to 100 scale [8,11].

The high prevalence of knee injuries was similar between males and females. Although not statistically significant the second-highest joint affected differed between males and females (ankle vs. hip). This sex difference may be related to biomechanical differences during running. Females compared to males have shown differences in kinematics in the transverse and frontal planes including greater hip adduction and internal rotation [35] and greater gluteus maximus activity on both level and inclined surface [36].

Similarly, illness did not differ between males and females. This finding is consistent with previous studies [37] that report no significant differences in upper respiratory symptoms (40% vs. 52%) during a 16-week winter training period in endurance athletes.

Some of the limitations of our study are the possible recall bias and self-diagnosis of injuries and illnesses by the athletes as a physician did not confirm the diagnosis. Regarding recall bias, it is possible that runners with non-time loss injuries had more difficulties remembering the characteristics of their injuries, even though the analysis time was short (4 weeks). In relation to the diagnosis of injuries and illnesses, classification was based on signs and symptoms. Despite this, self-reported injury questionnaires are accurate when compared to medical records, thus, they could be a good alternative for this type of study [38]. Although our definition of injury did not exclude those injuries generated by other activities different from running, only 11% of injured athletes reported an injury of a different origin. This is an important consideration, since most research analyzing injury prevalence in athletes define it as running-related injuries, without considering injuries from any other origin which may influence training or competition. Additionally, data such as weekly training hours during the last 4 weeks were not considered, since this measure was self-reported and instructions were not clearly understood by some participants. In some cases, participants indicated weekly kilometers and not hours. Possibly weekly distance would have been a better self-reported outcome. Objective measures of training load, such as the measurement of space-time parameters through the use of clocks with global positioning systems in ecological conditions [39], are recommended for future studies. The results should be interpreted with caution, because even though the sample did not have significant differences regarding the proportion by sex, age and competition distance, the response rate was low (55.2%), limiting the generalizability of our results.

## 5. Conclusions

Almost a third of athletes participating in a trail running event reported sustaining an injury and nearly a quarter contracting an illness in the 4 weeks before the competition. Gradual onset of injuries was the most common mechanism of injury and the knee the most frequently affected body region. Most athletes modified their training volume and/or intensity due to injury and most of the injured athletes perceived the injury would alter participation in the competition. Considering that a multidisciplinary team prepares these athletes at different phases of their training program, these results could help to generate education strategies and appropriate medical support before and during these competitions.

## Figures and Tables

**Figure 1 ijerph-18-05431-f001:**
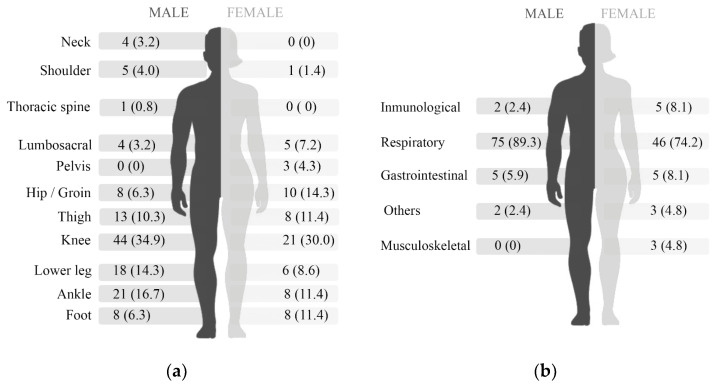
Anatomical region of (**a**) injury and (**b**) organ system affected by illnesses for males (values on the left) and females (values on the right).

**Table 1 ijerph-18-05431-t001:** Demographic and race distance characteristics of participants.

	Total *n* = 654	Injured Athletes *n* = 205	Non-Injured Athletes *n* = 449	*p*-Value	Ill Athletes *n* = 146	Non-Ill Athletes *n* = 508	*p*-Value
Sex ^‡^							
Female	241 (36.9)	74 (30.7)	167 (69.3)	0.787	59 (40.4)	182 (35.8)	0.312
Males	413 (63.1)	131 (31.7)	282 (68.3)		87 (59.6)	326 (64.2)	
Age (years) ^†^	36.2 [30.6–43.0]	36.7 [30.4–43.9]	35.9 [30.8–42.3]	0.595	36.4 [30.5–41.2]	36.1 [30.7–43.3]	0.632
Height (cm) ^†,§^	170.0 [163.0–175.0]	170.0 [163.0–175.0]	170.0 [163.0–175.3]		169.5 [162.3–176.0]	170.0 [164.0–175.0]	
Weight (kg) ^†,§§^	70.0 [60.0–78.0]	70.0 [60.0–77.8]	70.0 [60.0–79.0]		70.0 [60.0–80.0]	70.0 [60.0–78.0]	
BMI ^†,§§§^	24.0 [22.2–26.0]	24.1 [22.3–25.8]	23.9 [22.2–26.1]	0.996	24.2 [22.6–26.1]	23.9 [22.2–25.9]	0.343
Distance ^‡^							
11 k	194 (29.7)	64 (33.0)	130 (67.0)	0.411	49 (33.6)	145 (28.5)	0.817
18 k	136 (20.8)	36 (26.5)	100 (73.5)		27 (18.5)	109 (21.5)	
24 k	144 (22.0)	44 (30.6)	100 (69.4)		29 (19.9)	115 (22.6)	
45 k	101 (15.4)	36 (35.6)	65 (64.4)		24 (16.4)	77 (15.2)	
63 k	47 (7.2)	18 (38.3)	29 (61.7)		11 (7.5)	36 (7.1)	
Crossing (23 + 22 k)	32 (4.9)	7 (21.9)	25 (78.1)		6 (4.1)	26 (5.1)	

Note: *n*, number; cm, centimeters; kg, kilograms; BMI, body mass index; k, kilometers. ^†^ Median, Interquartile range [IQR]; ^‡^ absolute frequency, percentage (%); ^§^ missing data for 19 participants (5 females and 14 males; 4 injured and 15 non-injured athletes); ^§§^ missing data for 19 participants (4 females and 15 males; 3 injured and 16 non-injured athletes); ^§§§^ missing data for 22 participants (6 females and 16 males; 5 injured and 17 non-injured athletes).

**Table 2 ijerph-18-05431-t002:** Consequences on training and competition due to injury.

Consequences on Training	Total
Volume	*n* = 201 ^§^
Not affected	45 (22.4)
Mildly affected	45 (22.4)
Moderately affected	67 (33.3)
Significantly affected	25 (12.4)
Not able to train	19 (9.5)
Intensity	*n* = 198 ^§§^
Not affected	39 (19.7)
Mildly affected	50 (25.3)
Moderately affected	64 (32.3)
Significantly affected	30 (15.2)
Not able to train	15 (7.6)
Performance	*n* = 199 ^§§§^
Not affected	29 (14.6)
Mildly affected	63 (31.7)
Moderately affected	71 (35.7)
Significantly affected	27 (13.6)
Not able to train	9 (4.5)

Note: *n*, number; () percentage; ^§^ missing data for 4 participants (4 males); ^§§^ missing data for 7 participants (3 females and 4 males); and ^§§§^ missing data for 6 participants (2 females and 4 males).

## Data Availability

The data that support the findings of this study are available from the corresponding author, upon reasonable request.

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
