# Peer review of "Frequency of Injury and Illness in the Final 4 Weeks before a Trail Running Competition"

_ijerph, 2021, doi:10.3390/ijerph18105431_

Round 1
Reviewer 1 Report
Please check the following comments:
The paper " Frequency of injury and illness in the final 4 weeks before a 2 trail running competition” is very interesting, and the authors deserve great merit. This study determines self-reported injury and illness frequency in trail runners 4 weeks preceding competition; compare athletes with and without injury/illness by sex, age, body mass index (BMI), competition distance; describe mechanism of injury, anatomical region (injury)/organ system (illness) involved, consequences of injury on preparation, self-perception of in- 23 jury severity; compare anatomical region (injury) and organ system (illness) by sex. Before, I recommend it to be accepted, I need authors to address my comments and clarify some details. Meanwhile, I would like to congratulate the authors for the great work presented.
Title: Why was the study done in 4 weeks?
Abstract
Line 30-34: According to the definition of the problem in the two parts of injury and illness, please discuss the results of the two parts separately.
Introduction
Line 63: On what basis are illness and injury differentiated in this study?
Line 69: Please provide a better and more accurate definition of injury and illness in the introduction.
Line 91: Due to the separation of survey dimensions in the study, please examine the relationship of each variable with the survey dimensions briefly in the introduction.
Results:
In table 1, please re-write ill athletes (III athletes).
Discussion
Line 242: Why is the study of diseases only a report in itself? Can this method be cited due to its different levels?
Line 303: Given the above paragraph that the disease has been reported in the upper part of the respiratory and this report should be based on the doctor's opinion, in this line 303 and 304 there is no reason to compare self-report and medical tests.
Reviewer 2 Report
This is well presented paper on an interesting topic. The study used a cross-sectional design and the analyses are somewhat shallow. I have following comments to the authors.
- Please include a theoretical model on injury etiology as a basis for the analyses performed and reported, e.g. Bertelsen et al, 2017 (Bertelsen ML, Hulme A, Petersen J, Brund RK, Sørensen H, Finch CF, Parner ET, Nielsen RO. A framework for the etiology of running-related injuries. Scand J Med Sci Sports. 2017 Nov;27(11):1170-1180.) Use the model as a basis for the discussion.
- When was the survey returned, before or after the race? Had some runners already completed the race? Some of the participants reported that they were unable to participate in the competition due to injury or illness. How were these runners reached by the survey?
- What injuries or complaints did not result in time loss from sports? Please describe location and type. How did these injuries compare to time loss injuries with regard to association with perceptions of performance.
- Consider analysing a multipe model with perceived performance as endpoint variable and the different injury types as explanatory variables. See, for instrance. Tillander et al. 2019 (Gauffin H, Tillander B, Dahlström Ö, Lyth J, Raysmith B, Jacobsson J, Timpka T. Maintaining motivation and health among recreational runners: Panel study of factors associated with self-rated performance outcomes at competitions. J Sci Med Sport. 2019 Dec;22(12):1319-1323. )
- Please discuss possible recall bias in assocation with non-time loss injuries in the limitations section.
Reviewer 3 Report
Frequency of injury and illness in the final 4 weeks before a trail running competition
Journal: International Journal of Environmental Research and Public Health
The objective of this study was to: (i) determine self-reported injury and illness frequency in trail runners 4-weeks preceding competition; (ii) compare athletes with and without injury/illness by sex, age, body mass index (BMI), competition distance; (iii) describe mechanism of injury, anatomical region (injury)/organ system (illness) involved, consequences of injury on preparation, self-perception of injury severity; (iv) compare anatomical region (injury) and organ system (illness) by sex.
The approach of the study appears very original and the contents of the manuscript are quite interesting by his methodology and through the tools of quantification used.
The manuscript reads smoothly and is easy to understand. The aims, scope, and results of the study are clearly stated. I have very much enjoyed reading this paper. I find it interesting and clearly written, and satisfying also all the other publication criteria of the “International Journal of Environmental Research and Public Health”. The study provides a very valuable addition to this line of research, and adds relevantly to the subject with additional original findings. I thus find that this paper definitively delivers results that will surely be of interest to the readership of the “International Journal of Environmental Research and Public Health”. I recommend the publication of this interesting paper after the following; the authors need to refer to the works of Fourchet et al. and also the following ref: Biomechanics of Trail Running Performance: Quantification of Spatio-Temporal Parameters by Using Low Cost Sensors in Ecological Conditions by Noé Perrotin, Nicolas Gardan, Arnaud Lesprillier, Clément Le Goff, Jean-Marc Seigneur, Ellie Abdi, Borja Sanudo and Redha Taiar Appl. Sci. 2021, 11(5), 2093; https://doi.org/10.3390/app11052093.
Round 2
Reviewer 1 Report
The authors have made good changes to the version. Well done!! Before accepting, I ask the author to carefully consider the writing corrections throughout the article. There are many mistakes in this regard.
Throughout the text, it should be noted that the endpoint "." of the sentence should be presented after the reference. Sometimes commas "," are not used correctly.
Author Response
Dear editor, Thanks again for your comments. We have performed a careful review of our manuscript and made the requested changes. The changes we have made is in highlighted in track changes in the main document.